**METHOD**

# LocusMasterTE: integrating long-read RNA sequencing improves locus-specific quantification of transposable element expression

Sojung Lee[1,2], Jayne A. Barbour[1,2], Yee Man Tam[1], Haocheng Yang[1], Yuanhua Huang[1,3] and Jason W. H. Wong[1,2*]

*Correspondence:
jwhwong@hku.hk

[1] School of Biomedical Sciences,
Li Ka Shing Faculty of Medicine,
University of Hong Kong, Hong
Kong SAR, China
[2] Centre for Oncology
and Immunology, Hong Kong
Science Park, Hong Kong SAR,
China
[3] Center for Translational Stem
Cell Biology, Hong Kong Science
and Technology Park, Hong Kong
SAR, China

## Abstract

Transposable elements (TEs) can influence human diseases by disrupting genome integrity, yet their quantification has been challenging due to the repetitive nature of these sequences across the genome. We develop LocusMasterTE, a method that integrates long-read with short-read RNA-seq to increase the accuracy of TE expression quantification. By incorporating fractional transcript per million values from long-read sequencing data into an expectation–maximization algorithm, Locus-MasterTE reassigns multi-mapped reads, enhancing accuracy in short-read-based TE quantification. We validate the method with simulated and human datasets. LocusMasterTE may give new insights into TE functions through precise quantification.

**Keywords:** Transposable elements, Short-read RNA-seq quantification, Expectation–maximization model

## Background

Transposable elements (TEs) are repetitive genomic units accounting for half of the human genome [1]. TEs can also be grouped based on genetic composition and monophyletic origin: class > family > subfamily [1]. Consensus sequence within the subfamily accounts for the high similarity in sequence compositions of TEs [2]. In normal cells, TEs are firmly regulated through epigenetic control, including histone modifications, but in cancer cells, an extensive number of TEs are associated with epigenetic dysregulation [3]. With the accumulation of mutations through evolution [4], most TEs are now inactive at the transcriptional level in humans; however, recent studies have identified TEs with functional impact on health and disease [5]. Moreover, those remaining active TEs influence regulatory elements and can have substantial implications in tumorigenesis [6, 7].

Interest in the study of TEs has led to the development of tools for their quantification by short RNA-seq data. Some of these include Telescope [8], SalmonTE [9], SQuIRE [10], TEtranscripts [11], TEtools [12], and RepEnrich [13]. However, as TEs are highly repetitive, especially within the same subfamily, there are many cases where one read from short-read RNA-seq is aligned to numerous TEs with similar or even identical sequence composition [10]. In particular, newly inserted TEs (i.e., young TEs) generally accumulate fewer polymorphisms with less unique sequences, leading to high multi-mapping rates [11]. Some tools avoid the multi-mapping issue by only focusing on TE quantification at the subfamily level. However, considering individual TEs are under epigenetic regulation, not all TEs in the same subfamily will simultaneously be up or down-regulated [14]. Other tools have utilized the expectation–maximization (EM) algorithm to estimate TE expression at specific locus [12]. Nevertheless, there are still numerous instances where the EM algorithm can only partially resolve multi-mapping reads due to insufficient information.

In recent years, long-read sequencing has emerged as a complementary approach for analyzing genomes and transcriptomes. Due to its longer technical read length, long-read can increase the success rate in quantifying novel transcripts from repetitive regions and identifying full-length isoforms [15, 16]. However, its cost remains high. Hence, it is rarely applied to large datasets or at a depth comparable to short-read sequencing.

Taking advantage of the mappability of long-read and the depth of short-read RNA-seq data, we have developed LocusMasterTE, a long-read assisted short-read TE expression quantification method. LocusMasterTE integrates long-read information to aid the quantification of RNA expressed from individual copies of TEs in a locus-specific manner from short-read RNA-seq data. We demonstrate the benefits of LocusMasterTE over existing tools using simulated and human RNA-seq datasets.

## Results

### Long-read RNA-seq is superior to short-read RNA-seq at mapping young TEs

Long-read sequencing techniques can read fragments typically between 10 and 100 kilobases, and even up to several megabases [17]. This greatly decreases the proportion of multi-mapping reads during alignment to TE in transcriptomics data (chi-squared test, $p = 0.003891$, Fig. 1A). To examine the differences between long-read and short-read transcriptome sequencing for TE expression quantification, we used paired data generated from the human colon cancer cell line (HCT116) to compare transcript quantification at the TE subfamily and individual level (detailed in the Methods).

As the TE subfamily is an aggregation of individual copies, we observed good agreement between Oxford Nanopore Technologies (ONT) long-read and short-read (Spearman's rank correlation coefficient, $R^2 = 0.84$, $p < 2.2e - 16$, Fig. 1B); however, there was a more substantial difference at locus-specific copy level (Spearman's rank correlation coefficient, $R^2 = 0.32$, $p < 2.2e - 16$). For comparison, the correlation at coding genes is $R^2 = 0.57$ (Spearman's rank correlation coefficient, $p < 2.2e - 16$, Additional file 1: Fig S1A). To account for read depth, we downsampled the short-read sample to match the number of mapped bases and conducted the same comparisons. We consistently observed more conformity at the subfamily level compared with the locus-specific TE level (Additional file 1: Fig S1B).

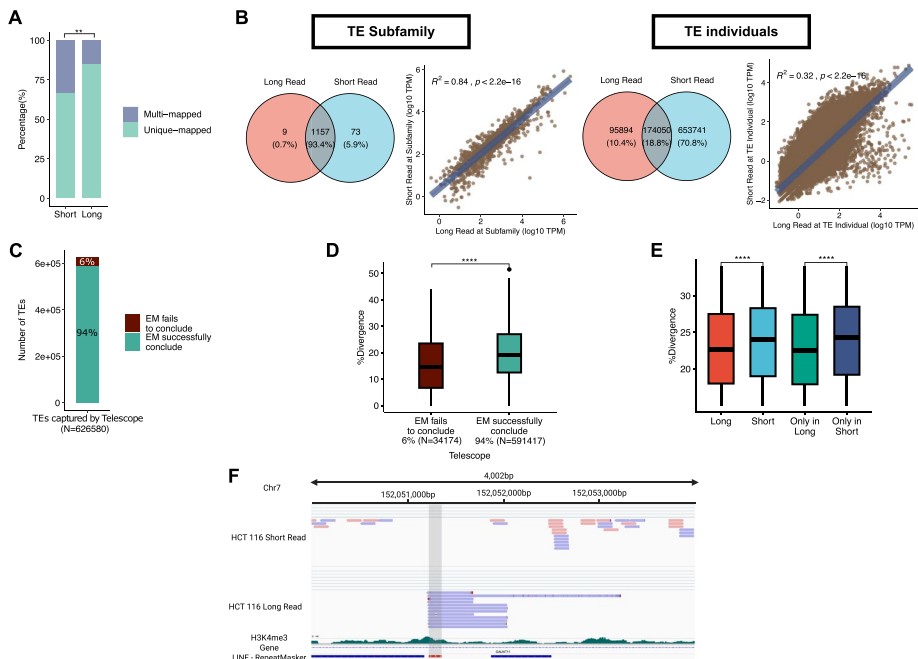

**Fig. 1** Comparison between long-read and short-read sequencing techniques. **A** Percentages for uniquely mapped reads and multi-mapped reads for short-read and long-read measured by BWA. Chi-squared test performed. **B** Figure showing at TE subfamily and TE individual levels. Left: Venn diagrams for long-read and short-read captured elements in two domains: TE subfamily and TE individuals. Right: Correlation plots between log10 TPM count for long-read and short-read in two same domains. **C** Overview of cases where EM takes part in short-read TE quantification by Telescope [8]. The total number of TEs corresponds to TEs initially aligned by Telescope [8]. Green: Expressed TEs that the EM model successfully identifies the origin of. Brown: Expressed TEs for which the EM model identifies more than two best hits, indicating an inability to determine the genomic origin of multi-mapped reads. **D** %divergence (% divergence from the consensus sequence) in EM has more than two best hits (brown). EM successfully concludes the best hit (green). A lower %divergence represents younger TE. Wilcoxon signed-rank test performed. **E** Box plot showing the difference between captured TEs based on sequencing types. Red: TEs captured in long-read. Light blue: TEs quantified by short-read. Green: TEs are exclusively captured in long-read (only in long). Dark blue: TEs are solely quantified by short-read (only in short). (ns: $p > 0.05$, $*: p \leq 0.05$, $**: p \leq 0.01$, $***: p \leq 0.001$, $****: p \leq 0.0001$). **F** IGV demonstrates a specific example of an advantage in long-read for TE study. The gray area represents reads for L1MC5a; chr7:152,051,230–152,051,364

TEs that have undergone transposition in the genome more recently (i.e., evolutionarily younger TEs) generally have fewer polymorphisms, rendering individual copies more challenging to distinguish and map uniquely. To correctly quantify young TEs using short-reads, TE quantification tools have applied the EM algorithm to reallocate multi-mapped reads based on an estimated probability. In theory, the EM algorithm should identify the most likely genomic locus for each multi-mapping read. However, we identified 6 % of cases where the EM algorithm could not identify a unique genomic location due to insufficient information (Fig. 1C). The estimated age (quantified by % divergence from a consensus ancestral sequence) of these ambiguously assigned TEs is significantly younger (i.e., lower % divergence) than other quantified (14.5% versus 19.0%; Wilcoxon signed rank test, $p < 2.2e - 16$, Fig. 1D), suggesting that existing EM-based methods are more prone to misquantify younger TEs.

Given that the longer read length in long-read sequencing leads to lower multi-mapping rates, we observed that the divergence of TEs quantified by long-read was

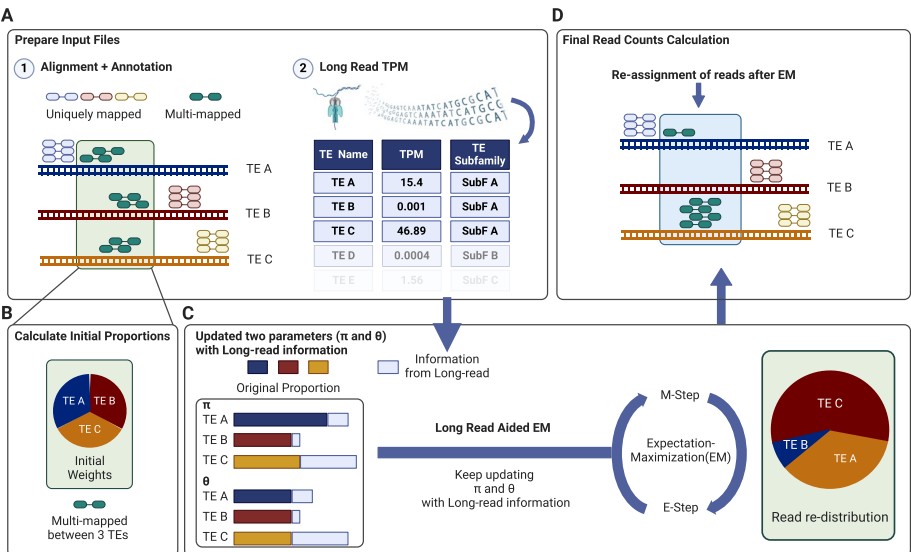

**Fig. 2** Schematic overview of LocusMasterTE. An example illustration for the LocusMasterTE pipeline is presented. **A** The short-read RNA sequencing data is aligned using the provided annotation file, and reads are classified into uniquely mapped and multi-mapped reads based on the alignment score. Uniquely mapped reads and multi-mapped reads are then aligned to TE A, TE B, and TE C. Long-read RNA sequencing, preferably ONT, is aligned using minimap2 [18], and TEs are quantified using featureCounts [19] with the same annotation file provided for short-read RNA sequencing. TPM counts for long-reads are calculated, and the belonging subfamily is provided for each individual TE. **B** Initially, equal weights are assigned to all three TEs—TE A, TE B, and TE C (TE A : 0.33, TE B: 0.33, TE C: 0.33). **C** Two critical parameters, pi ($\pi$) and theta ($\theta$), represent the proportion of fragments originating from the transcript and the proportion of non-unique fragments to be reassigned, respectively. These parameters are updated with TPM fractional values from long-read RNA sequencing. During the E-step, Maximum Likelihood Estimation (MLE) is calculated for all TEs. In the M-step, MAP estimates are updated. **D** Finally, after the EM algorithm coverages, best-aligned read counts are determined, leading to the final assignment of TE counts

lower than those captured by short-read (15.8% vs 18.7%; Wilcoxon signed rank test, $p < 2.2e-16$). The difference was increased when we compared a group of TEs exclusively quantified by long-read (median = 15.2%) and TEs only quantified by short-read (median = 19.5%; Wilcoxon signed rank test, $p < 2.2e-16$) (Fig. 1E). An example of a TE repetitive element mappable only by long-read is illustrated in Fig. 1F (L1MC5a; chr7:152,051,230–152,051,364).

## LocusMasterTE along-read assisted short-read TE quantification method

Given the benefits of long-read in studying TEs, we developed LocusMasterTE, which integrates long-read information to improve short-read-based TE quantification. As described above, the application of EM does not resolve all multi-mapping reads (Fig. 1C). LocusMasterTE additionally uses long-read information during the EM step, allowing improved resolution of multi-mapping (Fig. 2).

LocusMasterTE is divided into 4 stages: (1) alignment with annotations of reads to known transcripts, including TEs, (2) assigning initial proportions, (3) long-read aided EM model, and (4) final calculation of read counts. LocusMasterTE requires three inputs: a short-read sample in BAM format, a GTF file, and, most importantly, transcript per million (TPM) counts from a long-read sample. As LocusMasterTE was built

to quantify individual copies of TEs, a GTF containing the coordinates of individual TE copies is required. During the alignment and annotation stage, reads are divided based on the number of mapped TEs. Uniquely mapped read refers to a read fragment originating from one locus. When a read is aligned to multiple TE copies, the read is classified as a multi-mapped read (Fig. 2A). As with Telescope [8], LocusMasterTE represents the proportion of read fragments as a Bayesian mixture model and utilizes the EM algorithm to estimate parameters.

In reference to Telescope [8], LocusMasterTE uses two parameters, theta ($\theta$) and pi ($\pi$), for the EM model, each representing the proportion of fragments originating from the transcript and the proportion of non-unique fragments to be reassigned respectively (Fig. 2B). The main innovation of LocusMasterTE lies in the integration of long-read TPM fractions (details in the "Methods" section). For every EM iteration, long-read information is integrated into calculating the maximum a posteriori (MAP) for $\pi$ and $\theta$, and the updated MAPs are used for subsequent E-step (Fig. 2C). In order to demonstrate LocusMasterTE's capability to minimize cases where the EM algorithm is unable to identify a single genomic position for each read that maps to multiple locations, we conducted a similar analysis as depicted in Fig. 1C. The occurrences in which the EM algorithm failed to identify a single locus were reduced to 1% with the use of LocusMasterTE (Additional file 1: Fig S2), indicating the benefit of incorporating long-read RNA-seq information.

Although data from any long-read sample can be used in LocusMasterTE, the best accuracy would be achieved using matched cell or tissue types between long-read and short-read. Nevertheless, we recognized that long-read data may not always be available for the desired cell type, thus, LocusMasterTE introduces a weight parameter (*–long_read_weight*), allowing users to manage the contribution of long-read information. Lower weights are recommended with distant tissue types, but optimal weights vary between samples. We conducted tests with various weight (*–long_read_weight*) values from 0.01 to 1.5 on our in-house HCT116 short-read cell line sample along with a K562 long-read sample from the SG-NEx study [20]. Our findings revealed that values below 0.5 and above 1.0 (*–long_read_weight*) resulted in lower accuracy (Additional file 1: Fig S3). When an unmatched long-read sample is used, it is advisable to use a weight (*–long_read_weight*) smaller than 1.0. Otherwise, in most cases, a weight (*–long_read_weight*) of 1.0 is recommended (default). At the conclusion of the EM optimization, LocusMasterTE provides an updated alignment with finalized counts (Fig. 2D).

### Benchmarking LocusMasterTE on simulated data

To demonstrate the accuracy of LocusMasterTE, we simulated short-read RNA-seq datasets based on a long-read RNA-seq sample (details in the "Methods" section), enabling the use of the long-read data as the ground truth. Using the simulated short-reads, we compared the accuracy of LocusMasterTE for TE expression quantification with seven approaches: (1) unique counts (Fig. 3A), (2) featureCounts [19] (Fig. 3B), (3) RSEM [21] (Fig. 3C), (4) Telescope [8] (Fig. 3D), (5) SQuIRE [10] (Fig. 3E), (6) SalmonTE [9] (Fig. 3F), (7) LocusMasterTE (Fig. 3G). "Unique counts" was directly provided by Telescope [8]. For fair comparisons, salmonTE [9] was modified to output TE quantifies at individual loci, as it was originally designed to calculate TE at the subfamily

level. Moreover, we allowed mult-mapping read (-*M*) and overlapping features (-*O*) to be counted in featureCounts [19].

Prior to comparing TE counts, we tested the quality of simulated short-read using coding genes (Additional file 1: Fig. S4). Out of 19,120, 18,465 coding genes were consistently captured between long-read and simulated short-read (TP = 13,229, TN = 5,236, FP = 120, FN = 535). LocusMasterTE had no falsely detected TEs while having the highest number of correctly detected TEs (TP = 45,686). It had the highest precision among the seven approaches (precision: 1.0, recall: 0.86, *F*1 score: 0.92; Fig. 3H). FeatureCounts [19] showed relatively low precision (precision: 0.46, Fig. 3H). As featureCounts [19] reports all multi-mapping reads without the EM algorithm, low precision in featureCounts demonstrates the importance of the EM step when assigning multi-mapping reads. SQuIRE [10] captures a similar amount of TEs as LocusMasterTE, but has a higher false positive rate, leading to low precision (precision: 0.49, Fig. 3H). As LocusMasterTE was built upon Telescope [8], we carefully compared the performance of the two tools. LocusMasterTE achieved much higher TP and TN with lower FP and FN values, outperforming Telescope [8] (Fig. 3D, G). Based on this simulated dataset, LocusMasterTE is demonstrated to be the most accurate among the seven approaches.

To ensure compatibility with both ONT and PacBio Iso-Seq long-read RNA sequencing inputs, we conducted a performance comparison of LocusMasterTE using our simulated short-read RNA-seq from these two platforms. Since PacBio Iso-Seq primarily aims to identify novel transcripts, we utilized Talon [22] for quantification. While there is a slight variation in quantified TE expression between ONT and PacBio, we found that 94.7% (*N* = 85,467 of 97,746) of individual TEs quantified are consistently captured by both techniques (Additional file 1: Fig. S5A). Additionally, we observed a strong positive correlation between the two long-read techniques (Spearman's rank correlation coefficient, $R^2 = 0.9$, Additional file 1: Fig. S5B), indicating that both long-read approaches can serve as suitable inputs for LocusMasterTE.

### TEs uniquely quantified by LocusMasterTE are evolutionarily younger

To demonstrate the advantage of using LocusMasterTE to study TE expression, we used cell line samples with matched long and short-read RNA sequencing data as input for LocusMasterTE. This included one in-house dataset from HCT116 and five human cancer cell lines (A549, HCT116, HepG2, K562, and MCF-7) from the SG-Nex [20].

We examined TEs that showed expression changes between LocusMasterTE and Telescope [8] and found that these were generally evolutionarily younger TEs compared with those that were unaffected by the choice of method. This suggests that LocusMasterTE is indeed correcting the expression of younger TEs that are generally

(See figure on next page.)

**Fig. 3** Assessing the accuracy of LocusMasterTE in simulated short-read. 7 distinctive approaches to quantify TEs in a simulated short-read. Tested approaches are **A** unique counts provided by Telescope [8], **B** featureCounts [19], **C** RSEM [21], **D** Telescope [8], **E** SQuIRE [10], **F** salmonTE [9], and **G** LocusMasterTE. As a simulated short-read is generated from a long-read, all comparisons are conducted with a long-read TE quantification result. All quantified TE counts are converted to log10 TPM counts. For all approaches (**A**–**G**), true positive (TP), true negative (TN), false positive (FP), and false negative (FN) numbers are included. **H** Precision and recall values for seven approaches

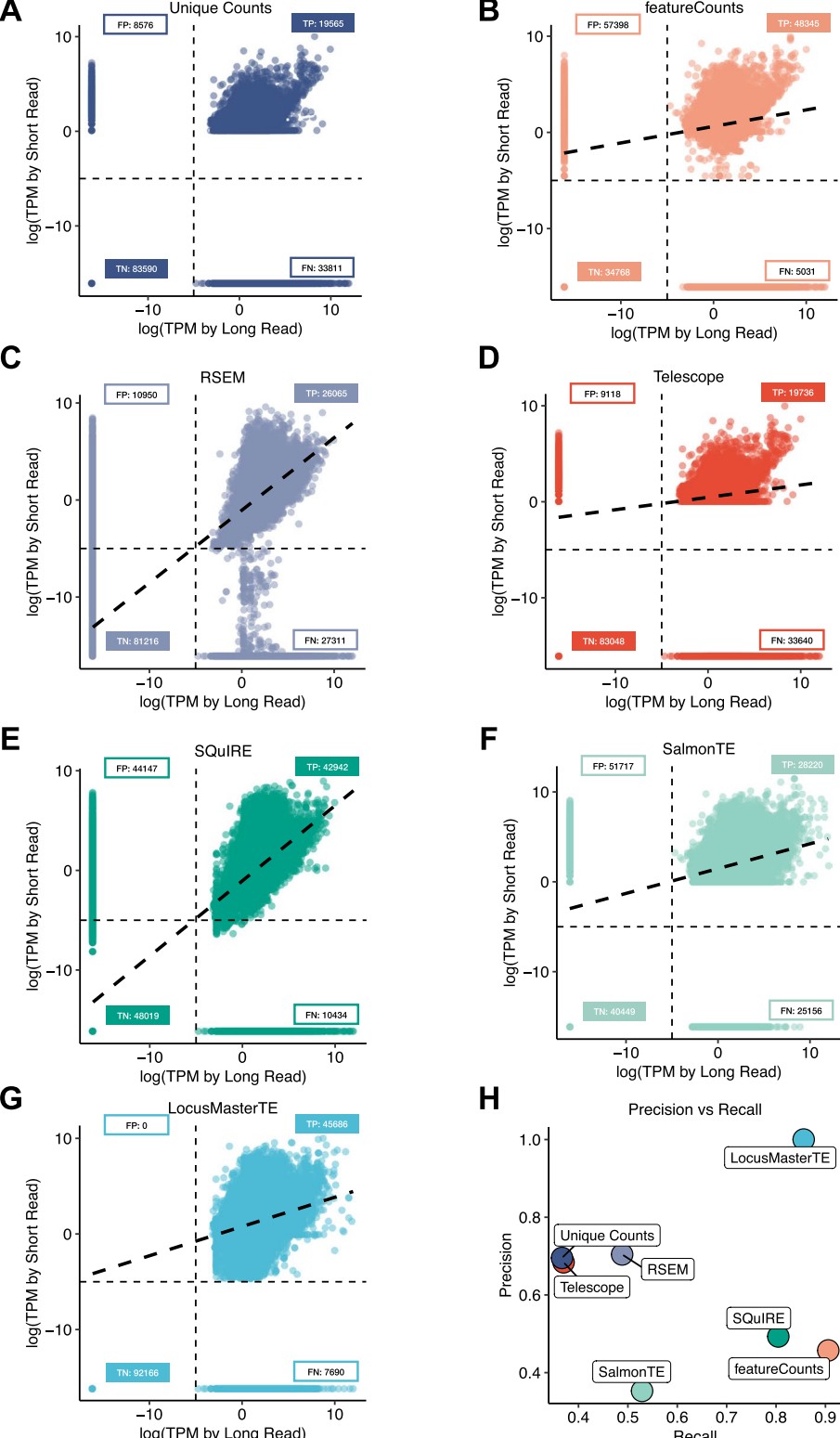

**Fig. 3** (See legend on previous page.)

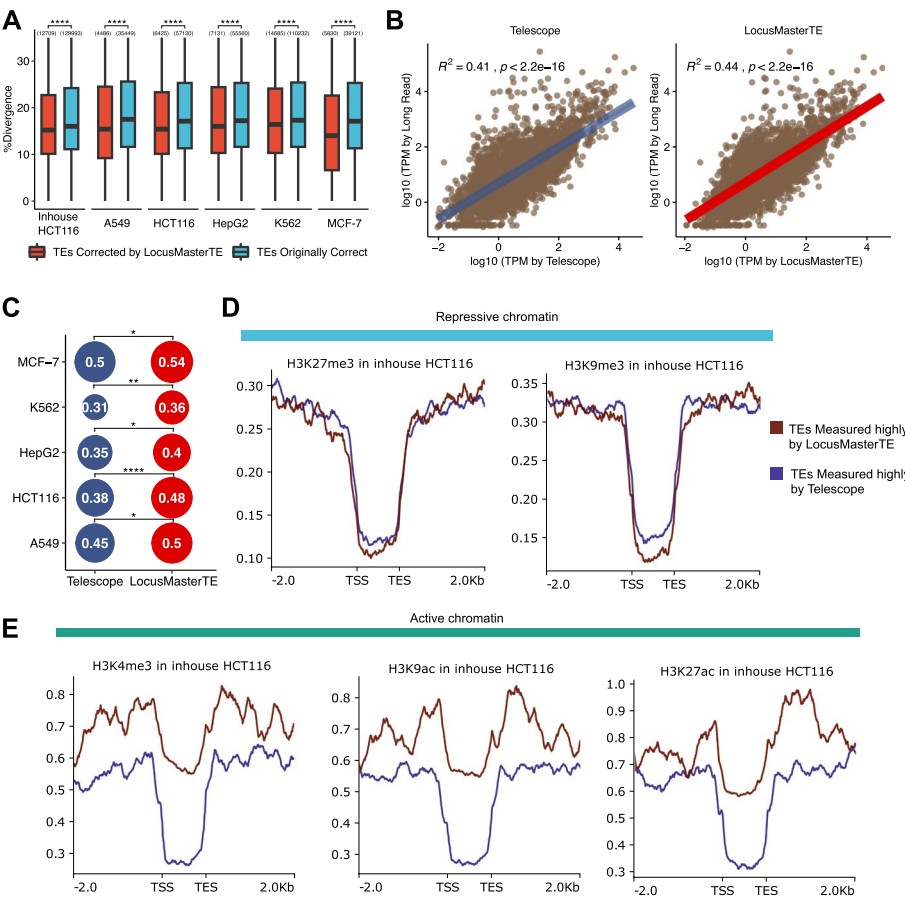

**Fig. 4** Young and active TEs identified by LocusMasterTE supported by histone markers. **A** Age distribution of TEs across all cell lines between two groups of TEs. Red: TEs quantified differently between LocusMasterTE and Telescope [8] (TEs corrected by LocusMasterTE). Blue: TEs quantified the same across LocusMasterTE and Telescope [8] (TEs originally correct). A paired *t*-test conducted for each cell line sample. Quantitative numbers are labeled above each box. **B** Spearman correlation plots for Telescope [8] and LocusMasterTE with long-read inhouse HCT116 cell line. log10 TPM counts are used. **C** Summary bubble plot for correlation with long-read in SG-NEx cell lines. Fisher r-to-z transformation was performed between two groups. Bonferroni correction also applied. (ns: $p > 0.05$, $*: p \leq 0.05$, $**: p \leq 0.01$, $***: p \leq 0.001$, $****: p \leq 0.0001$). **D** Two repressive chromatins fold change over control coverage of in-house HCT116 cell sample for the same two TE groups. Brown: TEs quantified higher in LocusMasterTE (TEs measured highly by LocusMasterTE). Blue: TEs are quantified higher in Telescope [8] (TEs are measured highly by Telescope). **E** Three active chromatins fold change over control coverage of in-house HCT116 cell line for two TE groups

difficult to quantify (Fig. 4A). TE counts generated by Telescope [8] and LocusMasterTE were then compared with long-read results to confirm the successful integration of long-read in cell line samples. We observed an increased correlation in TE expression with long-read results using LocusMasterTE (Spearman's rank correlation coefficient, $R^2 = 0.44$, $p < 2.2\mathrm{e}-16$) compared with Telescope (Spearman's rank correlation coefficient, $R^2 = 0.41$ $p < 2.2\mathrm{e}-16$) using the inhouse HCT116 dataset [8] (Fig. 4B). Due to the difference in average sequencing depth between in-house and SG-NEx samples (Additional file 1: Fig. S6), there were slight variations in correlation values with long-read. Nevertheless, the same trend was consistently observable in SG-NEx across all cell types. (Fisher r-to-z transformation, Bonferroni

correction, *p-adj* A549 = 0.0392, *p-adj* HepG2 = 0.0301, *p-adj* HCT116 = 0.0001, *p-adj* K562 = 0.0016, *p-adj* MCF-7 = 0.0314, Fig. 4C; Additional file 1: Fig. S7).

### TEs solely quantified by LocusMasterTE are marked by more active and less repressive chromatin

Histone modifications regulate gene expression, including those of TEs. We selected TEs that were differentially quantified by LocusMasterTE and Telescope [8], grouping them based on whether a TE had a higher expression value as quantified by the respective method (detailed in the "Methods" section). We then examined their associations with five histone marks: three active (H3K4me3, H3K9ac, H3K27ac) and two repressive (H3K27me3, H3K9me3). To avoid bias in the histone ChIP-seq data, we selected TE regions with a mappability score greater than 0.25, finding no significant difference between the two groups (median TEs with higher expression by Telescope) = 0.5006275 vs median (TEs with higher expression by LocusMasterTE) = 0.505051, $p = 0.1661$, Wilcoxon signed-rank test, Additional file 1: Fig. S8A). Between two groups of TEs, TEs highly captured by LocusMasterTE had less enrichment for the two repressive marks (Fig. 4D), and more enrichment for all three active marks (Fig. 4E) The same trends were observed across all cell lines (Additional file 1: Fig. S8B). This suggests that LocusMasterTE improves TE qualification as it is more consistent with independent epigenetic data.

### Presence of RNA editing validates novel transcripts identified by LocusMasterTE

To validate that reads mapping to TEs that LocusMasterTE reassigned are genuinely TE-derived, we utilized samples from the TCGA-COAD cohort to study RNA editing events at TEs. RNA editing refers to the post-transcriptional modification of RNA sequences by specific enzymes. Recent studies reveal that Alu elements account for more than 99% of human RNA editing events by forming double-stranded RNA that undergoes A-to-I editing [23]. The TCGA-COAD cohort ($N = 287$) was used to ensure sufficient sequence depth for RNA editing detection.

We derived the reference RNA editing trinucleotide signature from MDA5-protected Alu elements from 5-azacytidine-treated ADAR wild-type samples [23] and compared them with three groups from our analysis: (1) coding genes, (2) TEs detected by both Telescope [8] and LocusMasterTE, labeled as "common," and (3) TEs exclusively captured by LocusMasterTE, labeled as "only in LocusMasterTE." The RNA editing profile of TE classified as "common" and "only in LocusMasterTE" showed a similar pattern with the reference sites, with peaks at CTA > CCA, CTG > CCG, and GTG > GCG (Fig. 5A). This is supported by a high cosine similarity (Fig. 5B), but with a clear difference from coding genes that generally show much lower editing levels. This suggests that reads reassigned by LocusMasterTE are likely to be from TEs-derived RNA which are subjected to RNA editing.

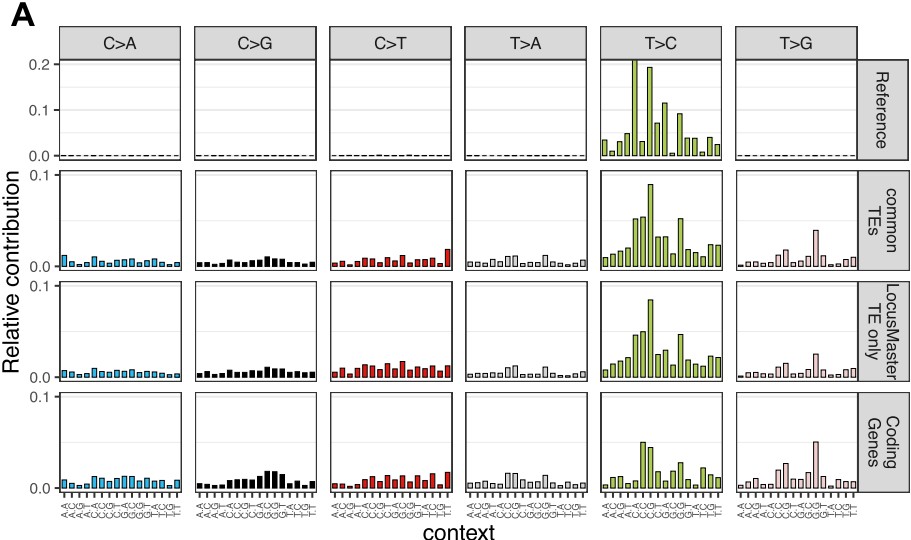

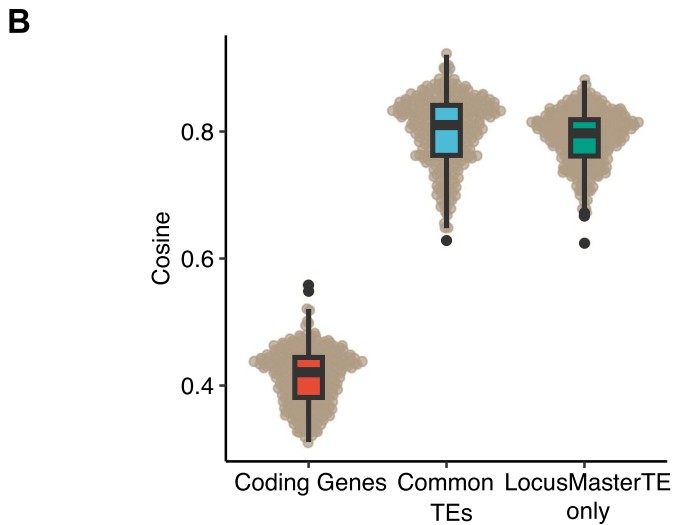

**Fig. 5** TEs solely captured by LocusMasterTE are active in RNA editing events. **A** Trinucleotide profiles for four groups: ADAR-WT-MDA5-5aza sample with inverted SINEs and ADAR1 [23], TEs commonly detected for both Telescope [8] and LocusMasterTE (common TEs), TEs solely detected by LocusMasterTE (LocusMasterTE only TEs), and coding genes (coding genes) (**B**) Cosine similarity values between "common TEs (blue)," "LocusMasterTE only TEs (green)," and "coding genes (red)" based on "Reference" for all TCGA-COAD tumor samples

## Discussion

Transposable elements play essential roles in cancer and complex biological diseases in humans [1]. However, due to the repetitive nature of TE sequences, it has been challenging to align and quantify TEs accurately. Since long-read has an advantage over short-read in mappability due to read length, we developed LocusMasterTE to integrate the long-read results using the EM algorithm. Including long-read results enables the EM algorithm to assign reads more correctly than conventional short-read TE quantification tools. In particular, LocusMasterTE resolves almost all cases where the read assignment remains ambiguous despite the use of the EM algorithm.

Also, LocusMasterTE allows users to use any long-read regardless of cell or tissue type.

The accuracy of LocusMasterTE was first illustrated using a simulated short-read dataset. Among seven TE quantification approaches, LocusMasterTE successfully achieved the highest recall and precision scores. Additionally, our study reveals the advantage of using LocusMasterTE in studying evolutionarily young TEs. As newly inserted TEs achieve fewer polymorphisms, many tools struggle to capture young elements correctly. Based on a comparison with Telescope [8], TEs uniquely quantified by LocusMasterTE were generally younger. TEs quantified by LocusMasterTE were also more consistent with active and repressive histone marks quantified independently by ChIP-seq [24]. Finally, we leverage the presence of RNA editing to validate that reassigned reads by LocusMaster TE are indeed TE-derived.

While we believe LocusMasterTE is robust and accurate for TE quantification, it does have a few limitations. Since LocusMasterTE relies on long-read data, having long-read sequencing information is essential. However, considering the higher cost of long-read sequencing compared to other techniques, not all samples have both short-read and long-read data simultaneously, or at least not with sufficient long-read sequencing depth. Although LocusMasterTE allows users to input long-read data from any type of sample, the best performance is achieved when using long-reads from a matched sample or cell type. When employing the same reference long-read sample and running LocusMasterTE on different short-read samples, the TE counts generated by LocusMasterTE may include some estimated counts. Secondly, the performance of LocusMasterTE depends on the coverage of long-read data. With deeper sequencing depth, LocusMasterTE quantifies TEs more accurately. Despite these limitations, LocusMasterTE achieves the most precise TE quantification among available tools and effectively identifies previously undetected active TEs.

## Conclusions

Leveraging the advantages of long-read sequencing, we developed LocusMasterTE, which incorporates long-read RNA-seq data as a prior term to resolve multi-mapped reads in short-read RNA-seq datasets. This approach is adaptable to both PacBio and ONT long-read inputs, allowing users to adjust the weighting of long-read information. Our analysis of cell lines demonstrated that LocusMasterTE effectively captures young TEs, with their activity supported by histone marks. Moreover, in the TCGA-COAD cohort, TEs identified by LocusMasterTE exhibited clear RNA editing signatures, highlighting its potential for uncovering novel insights into TE activities in patient samples.

## Methods

### Datasets used in study

We used three main datasets for our study: (1) Simulated short-read data from an in-house HCT116. (2) Paired short-read and long-read (ONT) RNA sequencing data from in-house HCT116 and five different cell line samples from the SG-Nex study [20]. (3) TCGA-COAD tumor samples: we utilized a total of 287 TCGA-COAD tumor samples. In-house HCT116 long-read was used as input for LocusMasterTE.

### Generation of HCT116 RNA sequencing data

The cells with the plate number HCT116 were cultured in McCoy's 5A media supplemented with 10% FBS and then seeded into 35 cm dishes. When they reached 80% confluence, RNA was extracted using the Qiagen RNeasy Mini kit (74,104), following the manufacturer's directions.

For short-read RNA sequencing, the samples were sent to Novogene Co., Ltd (Beijing) for library preparation and Illumina sequencing. The process involved enriching the mRNA with oligo (dT) beads, fragmenting it randomly, synthesizing cDNA from the fragmented mRNA using random hexamers as primers and then subjecting it to second-strand synthesis, end-pair, dA-tailing, and ligation of sequencing adapters. The sequencing was performed on an Illumina Novaseq 6000 instrument.

For long-read RNA sequencing, 50 ng of the RNA was used for library preparation using the Oxford Nanopore cDNA-PCR barcoding kit (Oxford Nanopore, SQK-PCB109), following the manufacturer's protocol assuming an average transcript length of 1.5 Kb. Then, 100 fmol of the library was loaded on one flow cell (Oxford Nanopore, FLO-MINSP6), which was sequenced on the MinION Sequencing Device (Oxford Nanopore, MIN-101B).

### Generation of simulated short-read RNA sequencing

To benchmark LocusMasterTE, we generated a simulated short-read from long-read RNA sequencing using the read simulator Spanki [25]. We used in-house HCT116 ONT as input for the simulation. The simulation was only performed on chromosomes 1–22, X, and Y. Spanki allows users to input a transcript coverage file, and we inputted HCT116 ONT long-read TPM values for each TE as coverage [25]. To match the number of bases in the generated FASTQ file for simulated short-read and long-read, we first obtained the simulated FASTQ file without mapping the number of bases. Let the number of bases mapped in long-read FASTQ as $\text{BaseMapped}_{long}$ and the number of bases mapped in simulated paired-end short-read FASTQ without mapping coverages as $\text{BaseMapped}_{sim.short}$. Then we calculated the ratio between the two as follows:

$$\frac{\text{BaseMapped}_{long}}{\text{BaseMapped}_{sim.short}} = \frac{22,77,300,856}{95,915,952} \approx 23$$

Based on the calculated values, we multiplied 23 in long-read TPM values and updated TPM values were used as coverage input for simulation. Default values 76 and 200 were used for read length and fragment size, respectively.

### Adding long-read information into the prior distribution of the EM algorithm

In this study, LocusMasterTE was developed based on the fragment reassignment mixture model, Telescope [8], to enable locus-specific quantification of transposable elements. Specifically, LocusMasterTE utilizes an expectation maximization (EM) algorithm [26] to calculate an optimal probability for each read assignment. The essential innovation of LocusMasterTE lies in the EM model, where long-read information is used as a prior term. Long-read sequencing data, ideally from a related cell

type, is first aligned to a reference genome such as hg38 using Minimap2 [18]. Following alignment, TE quantification of long-read is performed using featureCounts [19] with a GTF file, such as one generated by RepeatMasker [27], containing individual TE transcripts $T = \{t_1, t_2, \ldots, t_K\}$, where $K$ is the number of transcripts. Raw counts for each TE generated by featureCounts [19] is converted to transcripts per million (TPM), such that each transcript has a TPM value, $t_i^L$, (i.e., $T^L = \{t_1^L, t_2^L, \ldots, t_K^L\}$). To ensure compatibility during model integration, TPM counts are converted to fractions ($T^{Lf} = \left\{t_1^{Lf}, t_2^{Lf}, \ldots, t_K^{Lf}\right\}$). As sequence composition is very similar within subfamily, TEs are normalized within in subfamily.

$$t_i^{Lf} = \frac{t_i^L}{\sum_{i=1}^{K} t_i^L \in subT}$$

where *subT* is the set of TEs from the subfamily of $t_i$.

Subsequently, LocusMasterTE follows the conventional EM model with expectation (E-step) and maximization (M-step), where an integration of $t_i^{Lf}$ occurs. Finally, let a set of indicators $Y = \{y_1, y_2, \ldots y_N\}$.

$y_i = 1$ in multi-mapped cases
$y_i = 0$ otherwise

As introduced by Telescope [8], $\pi = \{\pi_1, \pi_2, \ldots, \pi_K\}$ and $\theta = \{\theta_1, \theta_2, \ldots, \theta_K\}$ are two sets of parameters: representing the proportion of observed fragments originating from each transcript, and similar reasignment parameters for multi-mapped reads, respectively. In the initialization stage, LocusMasterTE assigns equal weights to all reads. In E-step, LocusMasterTE calculates the posterior probability of partial assignment weights for all fragments, denoted as $E[t_i^{sa}]$ for the posterior probability of assigning read $S_a$ to transcript *i*. Next, in M-step, based on the estimated probability from the E-step, maximum a posteriori (MAP) is estimated for model parameters. Here, LocusMasterTE includes the $T^{Lf}$ term as the prior distribution. LocusMasterTE has an useful option, (*–long_read_weight*), allowing users to freely adjust the weights of the long-read during integration. Let *w* be the value for (*–long_read_weight*) option.

$$\hat{\pi}_i = \frac{\Sigma_{a=1}^{N} E[t_i^{sa}] + (w \cdot \sum_{i=1}^{K} \Sigma_{a=1}^{N} E[t_i^{sa}]) \cdot t_i^{Lf}}{\sum_{i=1}^{K} \Sigma_{a=1}^{N} E[t_i^{sa}] + \sum_{i=1}^{K}(w \cdot \sum_{i=1}^{K} \Sigma_{a=1}^{N} E[t_i^{sa}]) \cdot t_i^{Lf}}$$

$$\hat{\theta}_i = \frac{\Sigma_{a=1}^{N} E[t_i^{sa}] y_a + (w \cdot \Sigma_{a=1}^{N} y_a) \cdot t_i^{Lf}}{\Sigma_{a=1}^{N} y_a + \Sigma_{a=1}^{N}(w \cdot \Sigma_{a=1}^{N} y_a) \cdot t_i^{Lf}}$$

From the mixture model provided by Telescope [8], introducing a new parameter $T^{Lf}$ during the EM algorithm, changing the parameters $\pi$ and $\theta$, which consequently affects overall mixture weights in the model [8]. Using the updated MAPs of two parameters, LocusMasterTE runs E-steps and M-steps iteratively until the changes in two parameter estimates are less than the specific level, which is usually set to $\epsilon < 0.001$ [28]. All other formulas are in accordance with the Telescope [8].

**Other essential features in LocusMasterTE**

1. Due to differences in sequencing techniques, some TEs can only be detected by either short-read sequencing or long-read sequencing. However, because LocusMasterTE uses long-read TPM counts directly and updates parameters through multiplication, if the quantified TPM counts from long-read sequencing are 0, then regardless of how many TEs are captured by short-read sequencing, the final counts will also become 0. To address this issue and rescue TEs that are only quantified by short-read sequencing, we have introduced the (–*rescue_short*) argument. This argument replaces 0 with a small constant in the long-read TPM count. The default value is set to 0, but users have the flexibility to input any small constant (i.e., $1e^{-50}$) based on their preferences.

2. We have added an extra option named "long_read" in the (–reassign_mode) argument, originally introduced by Telescope [8]. "long_read" option re-distributes counts based on the weighted mean. Assume after running EM, $TE_A$, $TE_B$, and $TE_C$ are having same values $TE_A^c = TE_B^c = TE_C^c$ for read $R_1$. Then, the final count for $TE_A$ represented $TE_A^{fc}$, is formulated below.

$$TE_A^{fc} = \left( TE_A^c + TE_B^c + TE_C^c \right) \cdot \frac{t_A^L}{t_A^L + t_B^L + t_C^L}$$

**Gene and TE quantification**

We obtained the coding gene Gencode annotation of the hg38 version 42 Basic GTF file [29] from the UCSC table browser using MANE annotated gene filters [30]. The GTF file for TE quantification was created using RepeatMasker [27] for Genome Reference Consortium Human Build 38 (GRCh38) [31]. We used Minimap2 [18] with GRCh38 fasta to align ONT. For Pacbio Isoseq, we used the isoseq3 tool developed by PacificBiosciences. Specifically, *CCS* was used to compute circular consensus sequences, and *lima* and *isoseq refine* were applied to select and process *CCS*. Minimap2 [18] was used for alignment, and Talon [22] was used for quantification. For short-read alignment, we used STAR [32] as the aligner. TE quantification was performed using LocusMasterTE for all analyses, except for long-read and short-read comparisons in Fig. 1. We used the parameters below for LocusMasterTE.

- (–*prior_change*): both, meaning changing both $\pi$ and $\theta$
- (–*long_read_weight*): 1
- (–*rescue_short*): $1e^{-50}$
- (–*reassign_mode*): long_read

In Fig. 1, featureCounts [19] was used to assign genomic features to reads for both long-read and short-read. Additionally, since LocusMasterTE originated from Telescope, several comparisons were made between Telescope [8] and LocusMasterTE. For each TE, two values were generated: counts from Telescope [8] and counts from LocusMasterTE, $TE^{c.tel}$ and $TE^{c.loc}$, respectively. Throughout the study, transcripts per kilobase

million (TPM) was used instead of raw counts, with the sum of coding genes used for the TPM library size.

### Comparative analysis on proportion of multi-mapping reads

First, for short-read RNA-seq, the STAR aligner [32] was used to align the reads to the entire genome. After that, bedtools was used to filter out regions containing basic genes. Then, the remaining reads were remapped using BWA [33], and regions that overlapped with TEs were selected. For long-read RNA-seq, minimap2 [18] was initially used for the alignment. After that, unmapped reads were remapped using BWA, and reads corresponding to TEs were extracted. The number of multi-mapping reads and uniquely mapped reads was calculated based on the "FLAG: tag in the SAM file [34].

### Comparison with conventional tools

To compare the accuracy of LocusMasterTE, we quantified TEs in simulated short-read RNA sequencing using Telescope [8], RSEM[21], SQuIRE [10], FeatureCounts [19], and SalmonTE [9]. To generate an index for SalmonTE, we used a fasta file consisting of TE individual sequences, which was generated by a locus-specific TE expression study [14]. We used STAR for LocusMasterTE and other tools like Telescope [8], FeatureCounts [19], and SQuIRE [10]. Bowtie2 [35] was used as an aligner for RSEM [21]. As the simulated short-read was generated from a long-read RNA sequencing sample, we considered quantified TE counts from the long-read as the ground truth. We only considered TEs captured by at least one approach, removing unsuccessfully simulated TEs. We calculated accuracy and recall values compared with the ground truth based on the confusion matrix generated for each tool.

### Age analysis in cell-line samples

We acquired % divergence (% divergence from ancestral sequences) of TE individuals from the RepeatMasker table at the UCSC database.

TEs were divided into two groups. TEs having the same counts between Telescope [8] and LocusMasterTE were considered as "originally correctly measured TEs." For TEs quantified differently between Telescope [8] and LocusMasterTE were defined as "corrected by LocusMasterTE," because those TEs were either over-estimated or under-estimated in Telescope [8].

- $TE^{c.loc} = TE^{c.tel}$: TEs quantified the same in both tools
- $TE^{c.loc} \neq TE^{c.tel:}$: TEs quantified differently

Student's *T*-test was conducted between two groups.

### Histone marker analysis in cell-line

Five histone modifications representing active (H3K4me3, H3K27ac, and H3K9ac) and inactive (H3K27me3 and H3K9me1) chromatin were used for histone analysis. K50 umap bigwig [36] was used to extract mappability over TE regions. Considering the presence of mappability bias between the two groups, TE regions with mappability

higher than 0.25 were selected. Fold change over control BigWig files from ChIPseq data of the five histone marks for each cell line were downloaded from the ENCODE project [24]. TEs were re-divided into two groups. BigWigAverageOverBed was used to calculate the average fold change over the control signal in two groups of TEs.

- $TE^{c.loc} > TE^{c.tel}$: TEs having higher expression in LocusMasterTE
- $TE^{c.loc} < TE^{c.tel}$: TEs having higher expression in Telescope [8]

For better comparison, TEs with non-trivial differences between Telescope [8] and LocusMasterTE $\left| TE^{c.loc} - TE^{c.tel} \right| > 1$ were selected. Then, we used computeMatrix from deepTools [37] to calculate scores.

### RNA editing events in TCGA-COAD

TEs were first grouped into two as follows:

1. $TE^{c.loc} \neq 0 \land TE^{c.tel} = 0$: TEs exclusively measured by LocusMasterTE
2. $TE^{c.loc} \neq 0 \land TE^{c.tel} \neq 0$: TEs both measured by two tools

A total of 35,000 TEs were randomly selected for the two groups. In addition, 35,000 coding genes were chosen for comparison. The ADAR-WT-MDA5-5aza sample from an epigenetic therapy study in inverted SINEs and ADAR1 [23] was used as the "reference" sample to extract expected editing signatures. The same GRCh38 fasta [31] and genomic location files mentioned above were used, and bam-readcount [38] was employed to gather information at specific nucleotide positions. Germline SNPs overlapping with the gnomAD database [39] were filtered out using bcftools [40]. Alu regions were used for all groups except for the coding genes group. The R package MutationalPatterns [41] was used to illustrate mutational profiles for all groups in trinucleotide contexts. Additionally, we conducted a series of cosine similarity tests using the *lsa* R package [42] to compare all groups with the TCGA-COAD samples [43].

### Supplementary Information

---

Additional file 1. Supplementary figures of LocusMasterTE: integrating long-read RNA sequencing improves locus-specific quantification of transposable element expression file contains Fig. S1 – S8.

Additional file 2. Peer review history

---

**Review history**
The review history is available as Additional file 2.

**Peer review information**

**Authors' contributions**
Sojung Lee: Conceptualization, Data curation, Software, Formal analysis, Investigation, Methodology, Resources, Validation, Visualization, and Writing - original draft, review & editing. Jayne A. Barbour: Investigation. Yee Man Tam: Investigation. Haocheng Yang: Investigation. Yuanhua Huang: Methodology, Writing - review & editing. Jason W.H. Wong: Conceptualization, Funding acquisition, Methodology, Resources, Project administration, Supervision, Writing - original draft, review & editing.

## Funding

This work was supported by the Centre for Oncology and Immunology under the Health@InnoHK Initiative funded by the Innovation and Technology Commission, the Government of Hong Kong SAR, China, and the National Key R&D Program of China (2022YFE0200100) (J.W.H.W).

## Data availability

Raw RNA sequencing datasets for the TCGA-COAD study can be downloaded from the TCGA data portal [https://portal.gdc.cancer.gov/] [44]. Our study used processed gene count data from TCGA-COAD RNA sequencing, which can be found on the UCSC Xena Hub [https://gdc-hub.s3.us-east-1.amazonaws.com/download/TCGA-COAD.star_counts.tsv.gz] [45]. Histone datasets and HCT116 pacbio isoseq data are retrieved from ENCODE [https://www.encodeproject.org/]. Short-read and ONT RNA sequencing datasets for SG-NEx are available through their GitHub [https://github.com/GoekeLab/sg-nex-data]. Inhouse HCT116 long-read and short-read are deposited at Gene Expression Omnibus (GEO) [https://www.ncbi.nlm.nih.gov/geo)] under GSE225377 and GSE225380 respectively. We analyzed the ADAR-WT-MDA5-5aza sample from the accession code GSE145639 [23].

LocusMasterTE [46] is an open-source tool licensed under MIT. The dataset used for analysis in the paper is deposited in Figshare [47]. The code for LocusMasterTE is accessible through its GitHub repository at https://github.com/jasonwong-lab/LocusMasterTE. Scripts used to generate the main figures and supplementary information files are also available at the LocusMasterTE paper's GitHub repository: https://github.com/jasonwong-lab/LocusMasterTE_paper and Zenodo [46]. Detailed sample IDs and accession numbers for the ENCODE and SG-NEx datasets can be located in the Data_lists folder of the same repository.

# Declarations

### Ethics approval and consent to participate

Ethics approval was not required for this study as it did not involve human or animal subjects; therefore, these regulations are not applicable.

### Competing interests

All authors have no competing interests to declare.

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

## 