## [Additional file 2. Peer review history · Genome Biology]

Review history

First round of review

Reviewer 1

review of the manuscript entitled "Integration of long-read RNA sequencing improve locus-specific quantification of transposable elements" by Lee et al..

This manuscript presents new software, LocusMasterTE that integrates long read and short read sequencing data in order to optimize the quantification of locus-specific TE expression.

I found the manuscript very hard to read because of the poor quality of written english. I should admit that I could not understand some parts of the results section. In addition, the structure of the paper is a bit awkward with a lot of repetitions in the introduction and the results section, with insufficient information in the methods section and mixtures of matters in a single paragraph. Overall, this manuscript lacks the necessary focus to be published in GB.

1- On the software itself : the authors claim that they have considerably improved TELESCOPE, a software designed to quantify repeats expression, by including long read sequencing data. The results of the analyses provided by the authors on benchmarking are convincing suggesting that LocusMasterTE may indeed help resolve locus-specific TE transcriptional activity. However, the authors do not mention what type of long read sequencing technology was used : Nanopore or PacBio. One would expect to get an exhaustive comparison of both in terms of the software performance.

2- 2.4 results section. The authors mix two parameters in a single paragraph : the age of the insertions and the chromatin state. As a result, the reader can not understand their impact on the software's performance.

3- 2.5 results section. "Biological connection to tissue samples" The title does not make sense, but one understands that the authors will try to use their software to detect locus, the transcriptional variation of which are linked to carcinogenic activity of tissues. This is the case for genes, but in the same paragraph, the authors use the software to detect new associations. They conclude that it is superior to any existing software, but at this point, one misses some wetlab validation of these results. This is confusing. The last part on "survival data" is the most unfocused of the paper. Apparently, the authors have used their software to identify TE insertions that may be correlated with higher survival rate. Again, they reach the conclusion that this tool could help resolve some SVs associated with cancer "resistance", but no data is provided on the type of genes that has been identified, if the results make any sense at biological level etc...

Below are some examples of sentences that make the manuscript hardly readable :

... which paves a crucial step to understanding TE mechanisms in cancer...

... the EM model should converges with one optimal best hit....

... to understand the distinctive characteristics of LocusMasterTE ...

... As TEs impose great influences on nearby genes...

... we utilized LocusMasterTE to observe the connection in the survival data of TCGA samples...

... As newly inserted TEs achieve fewer polymorphisms, many tools struggle to capture young elements correctly...

Reviewer 2

In this manuscript, the authors propose an amelioration of the model implemented in the tool Telescope to incorporate information from long-read transcriptomic data allowing a better estimate of transposable element (TE) expression measure. This approach seems interesting but the method section is not very clear and should probably be reorganized to help follow what the authors have done. I have detailed below my main concerns.

Major points

1) In the section 2.1 of the results, where the authors compare long-reads vs short-reads, it should be clearly indicated (and in the figures) on what data it was done. Globally, it is never clearly mentioned that all the analyses have been performed on human data, especially in the abstract.

2) in the same section, about the median age of the TE, I do not understand how the age can be expressed in bp? Age of sequence can usually be estimated when comparing two sequences to compute their % divergence; it is also possible to use more elaborated models to take evolutionary parameters into account like the Kimura distance. Moreover, the statistical test used is not indicated. Similarly, the figure 1D should state %divergence in ordinate, or the Kimura distance.

3) section 2.2 : you should talk about « TE copies » or « TE insertions » not « TE transcripts » (same in the method section).

4) The Figure 2 is important since it allows to visually explain the approach. However, it is a little bit difficult to follow. On the panel A, the representation of unique and multimapped reads is misleading. It looks like unique reads are also multimapped. Moreover the multimapping reads are not really pointed out (at least one read of a given pair should be assigned to several copies). The parameters, theta and pi should also be represented in the figure. The panels B and C are also not clear. I do not understand to what correspond "Read 1" for example.

5) Supplementary Figure 3 is missing in the main manuscripts, there is only the title and the legend (p46).

6) in section 2.4, again, concerning the TE age, it would be better to have %divergence in addition to the statistical test names and associated p-values (that should be corrected for multiple testing). It should also be mentioned in the text that fig4B has been done using HCT1116 data. I am also wondering why the R2 for this tissue is different in the figure 4E.

The increase in correlation is not very high. It is possible to estimate the added-value by comparing both trends statistically? It seems less efficient than compared to the simulation.

7) I am quite puzzled by the results in section 2.5. What about polymorphic TE insertions? All RNA-seq mappings are done on a reference genome, with a particular set of TE insertions. However the RNAs come from other individuals which will have different TE insertions, especially the most recent ones. Then, I am wondering how this may affect the results. Some of the reads (either short or long) will come from insertions that may not be in the reference genome. I am not sure what the authors want to show here. The TE expression correlation and correlation of gene expression do not prove anything.

8) The method section is not very clearly organized. I would suggest a complete reorganization that would

help the reader follow exactly what has been done. Moreover, some titles are very not informative (like for example the section 4.6).

9) about the option concerning the weight that can be associated to long-reads in the model. I think this deserves to be better explained, especially, what are the possible choices of the user, what they mean, what are their limits etc. It will be difficult to always find both short and long reads obtained from the same dataset which may really be a problem. I would like to have more information and test about what is happening if both data are not exactly coming from the same source.

10) section 4.4, p19, line 9: simple repeats are not transposable elements. There is no real point to mention them.

11) section 4.4: the age analysis needs to be performed differently. It is not clear what is the "milliDiv" column present in the repeatmasker file (not database). And what file exactly. I do not recall that this column exists in the .out output file. As I specified before, age should be estimated as %divergence or using Kimura distance metrics.

12) in section 4.10.2 about histone enrichment analysis. This section should be more detailed, explaining the type of information that can be found in bigwig files, what scores were calculated exactly and how it is possible to reconcile results from chip-seq that comes from short-reads with the same multi-mapping problems as RNA-seq short-read when it comes to TEs.

13) it would be nice to have the same figure as figure 1C with the results provided by LocusMaster to see how the % of cases where the EM is not able to assess unique matches evolves.

Minor points

1) P3 : the « Endogenous » term should be removed since it is redundant with the term Transposable elements which are by definition endogenous.

2) there is a repeat in the second sentence of the introduction p3 (« Based on genetic etc. »).

3) the authors indicate that "most TEs are now inactive at the transcriptional level" (p3, line25). But it actually depends on the species considered. They should precise that it is the case in the human genome.

4) p6 line 51 coordinations => coordinates

5) p13 line 8: TEs do not "impose" essential roles. I would modify this sentence.

6) there are many typos (too numerous to be pointed out individually) in the whole manuscript that need to be corrected

Response to Reviewer comments

Reviewer: 1

1. On the software itself : the authors claim that they have considerably improved TELESCOPE, a software designed to quantify repeats expression, by including long read sequencing data. The results of the analyses provided by the authors on benchmarking are convincing suggesting that LocusMasterTE may indeed help resolve locus-specific TE transcriptional activity. However, the authors do not mention what type of long read sequencing technology was used : Nanopore or PacBio. One would expect to get an exhaustive comparison of both in terms of the software performance.

We appreciate the reviewer's comment. Initially, we tested LocusMasterTE with ONT technology only. In response to the suggestion, we have now also tested LocusMasterTE using PacBio long read RNA isoseq data. PacBio isoseq is primarily designed to discover novel transcripts and is not widely used for transcript quantification. Nevertheless, a method called Talon, is recently described (currently on BioRxiv) to enable RNA quantification from PacBio SMRT sequencing. We tested on simulated short-read RNA-seq. ONT and Pacbio long reads gave comparable results quantitative results for TEs at the individual level (spearman correlation estimates (ρ) = 0.9). We have now added a section to the results as follows:

Supplementary Fig 5B. Quantified TE expression between ONT vs PacBio on simulated short-read

Page 5, line 163-172:

To ensure compatibility with both ONT and PacBio Iso-Seq long-read RNA sequencing inputs, we conducted a performance comparison of LocusMasterTE using our simulated short-read RNA-seq from these two platforms. Since PacBio Iso-Seq primarily aims to identify novel transcripts, we utilized Talon [22] for quantification. While there is a slight variation in quantified TE expression between ONT and PacBio, we found that 94.7% (N=85467 of 97746) of individual TEs quantified are consistently captured by both techniques (Supplementary Fig 5A). Additionally, we observed a strong positive correlation between the two long-read techniques (spearman correlation estimates (ρ) =

0.9, Supplementary Fig. 5B), indicating that both long-read approaches can serve as suitable inputs for LocusMasterTE.

2. **2.4 results section.** The authors mix two parameters in a single paragraph : the age of the insertions and the chromatin state. As a result, the reader can not understand their impact on the software's performance.

We thank the reviewer for this comment. We agree that it may be clearer to separately describe the benefit of using LocusMasterTE on improving the ability to quantify younger TEs and its ability to quantify expressed TEs defined by increased active and decreased repressive chromatin in two different sections. Our original section 2.4 (now does not have section number) is now separated into two sections with the following subtitles:

TEs uniquely quantified by LocusMasterTE are evolutionarily younger

TEs solely quantified by LocusMasterTE are marked by more active and less repressive chromatin

3. **2.5 results section.** "Biological connection to tissue samples" The title does not make sense, but one understands that the authors will try to use their software to detect locus, the transcriptional variation of which are linked to carcinogenic activity of tissues. This is the case for genes, but in the same paragraph, the authors use the software to detect new associations. They conclude that it is superior to any existing software, but at this point, one misses some wetlab validation of these results. THIS is confusing. The last part on "survival data" is the most unfocused of the paper. Apparently, the authors have used their software to identify TE insertions that may be correlated with higher survival rate. Again, they reach the conclusion that this tool could help resolve some SVs associated with cancer "resistance", but no data is provided on the type of genes that has been identified, if the results make any sense at biological level etc...

We thank the reviewer for pointing out suggestions. We agree that our description of the benefit of applying LocusMasterTE to cancer cohorts may be unfocused. The editor has also suggested that we revised our manuscript as a Methods paper. Thus, we have now removed, and only use the TCGA-COAD data to help us demonstrate that the level of RNA editing of TE transcripts uniquely identified by LocusMasterTE is comparable to other commonly identified TE transcripts. This helps reinforce that these TEs quantified by LocusMasterTE are likely to be authentic. The following changes were made to the manuscript:

First, we change the title of original section 2.5 (now does not have section number) and revised the section to only describe the RNA editing results.

Page 6-7, line 207-225:

Presence of RNA editing validates novel transcripts identified by LocusMasterTE

To validate that reads mapping to TEs that LocusMasterTE reassigned are genuinely TE- derived, we utilized samples from the TCGA-COAD cohort to study RNA editing

events at TEs. RNA editing refers to the post-transcriptional modification of RNA sequences by specific enzymes. Recent studies reveal that Alu elements account for more than 99% of human RNA editing events by forming double-stranded RNA that undergoes A-to-I editing [22, 23]. The TCGA-COAD cohort (N = 287) was used to ensure sufficient sequence depth for RNA editing detection. We derived the reference RNA editing trinucleotide signature

from MDA5-protected Alu elements from 5-azacytidine-treated ADAR wild-type samples

[28] and compared them with three groups from our analysis: (1) coding genes, (2) TEs detected by both Telescope [8] and LocusMasterTE, labeled as "common", and (3) TEs exclusively captured by LocusMasterTE, labeled as "only in LocusMasterTE". The RNA editing profile of TE classified as "common" and "only in LocusMasterTE" showed a similar pattern with the reference sites, with peaks at CTA > CCA, CTG > CCG, and GTG > GCG (Fig. 5A). This is supported by a high cosine similarity (Fig. 5B), but with a clear difference from coding genes that generally show much lower editing levels. This suggests that reads reassigned by LocusMasterTE are likely to be from TEs-derived RNA which are subjected to RNA editing.

4. Below are some examples of sentences that make the manuscript hardly readable :
- ... which paves a crucial step to understanding TE mechanisms in cancer...
 - ... the EM model should converges with one optimal best hit....
 - ... to understand the distinctive characteristics of LocusMasterTE ...
 - ... As TEs impose great influences on nearby genes...
 - ... we utilized LocusMasterTE to observe the connection in the survival data of TCGA samples...
 - ... ` ...

We appreciate the reviewer for this comment. We thoroughly revised our manuscript. All previously mentioned sentences have been either removed or replaced with clearer ones. Here are a few examples:

Page 3, line 79-80:

In theory, EM algorithm should identify the most likely genomic locus for each multi- mapping read.

Page 5, line 174-176:

To demonstrate the advantage of using LocusMasterTE to study TE expression, we used cell line samples with matched long and short-read RNA sequencing data as input for LocusMasterTE.

Reviewer: 2

1. In the section 2.1 of the results, where the authors compare long-reads vs short-reads, it should be clearly indicated (and in the figures) on what data it was done. Globally, it is never clearly mentioned that all the analyses have been performed on human data, especially in the abstract.

We thank the reviewer for this suggestion. All of the analyses were performed on data from human cell lines or TCGA-COAD samples. We have now described this more clearly in the abstract and other sections of the manuscript as follows:

Page 2, line 62-66:

To examine the differences between long-read and short-read transcriptome sequencing for TE expression quantification, we used paired data generated from the human colon cancer cell line (HCT116) to compare transcript quantification at the TE subfamily and individual level (detailed in the Methods).

Additionally, we mentioned in the abstract that the data used are all human.

Abstract:

Transposable elements (TEs) can influence human diseases by disrupting genome integrity, yet their quantification has been challenging due to the repetitive nature of these sequences across the genome. We developed LocusMasterTE, a method that integrates long-read with short-read RNA-seq to increase the accuracy of TE expression quantification. By incorporating fractional transcript per million (TPM) values from long-read sequencing data into an expectation-maximization (EM) algorithm, LocusMasterTE reassigns multi-mapped reads, enhancing accuracy in short-read-based TE quantification. Validated with simulated and human datasets, LocusMasterTE holds promise for revealing new insights into TE functions through precise quantification.

2. in the same section, about the median age of the TE, I do not understand how the age can be expressed in bp? Age of sequence can usually be estimated when comparing two sequences to compute their % divergence; it is also possible to use more elaborated models to take evolutionary parameters into account like the Kimura distance. Moreover, the statistical test used is not indicated. Similarly, the figure 1D should state %divergence in ordinate, or the Kimura distance.

We thank the reviewer for this correction. We used %divergence from the repeat masker track, representing the divergence from the consensus sequence, where a lower %divergence represents younger TEs. We added an explanation of %divergence in the legend of figure 1 and text.

Fig 1E. %divergence between TEs captured in Long (Nanopore) vs Short RNA-seq in in-house HCT116.

Page 3, line 82-86:

The estimated age (quantified by % divergence from a consensus ancestral sequence) of these ambiguously assigned TEs is significantly younger (i.e. lower % divergence) than other quantified (14.5% versus 19.0%; p-value < 0.0001, Student's t-test, Fig. 1D), suggesting that existing EM-based methods are more prone to misquantify younger TEs.

3. **section 2.2 : you should talk about « TE copies » or « TE insertions » not « TE transcripts » (same in the method section).**

We appreciate the reviewer for bringing this up. In the original manuscript, we agree that we did not clarify the definition of “TE transcripts”. As LocusMasterTE quantifies TEs in RNA-sequencing data, “TE transcripts” refer to RNA being expressed by different copies of TEs across the genome. We feel that if we definite our meaning of TE transcripts more clear, it would be better to retain this rather than changing to TE copies or TE insertions which are more easily associated with describing TEs at the DNA level. We further clarify the TE transcript in our introduction.

Page 2, line 52-56:

Taking advantage of the mappability of long-read and the depth of short-read RNA-seq data, we have developed LocusMasterTE, a long-read assisted short-read TE expression quantification method. LocusMasterTE integrates long-read information to aid the quantification of RNA expressed from individual copies of TEs in a locus-specific manner from short-read RNA-seq data.

Page 8, line 315-318:

Following alignment, TE quantification of long-read is performed using featureCounts [18] with a GTF file, such as one generated by RepeatMasker [24], containing individual TE transcripts $T=\{t_1, t_2, \dots, t_K\}$, where K is the number of transcripts.

4. **The Figure 2 is important since it allows to visually explain the approach. However, it is a**

little bit difficult to follow. On the panel A, the representation of unique and multimapped reads is misleading. It looks like unique reads are also multimapped. Moreover the multimapping reads

are not really pointed out (at least one read of a given pair should be assigned to several copies). The parameters, theta and pi should also be represented in the figure. The panels B and C are also not clear. I do not understand to what correspond "Read 1" for example.

We are thankful to the reviewer for the feedback. We have made the necessary revisions to Figure 2 as suggested. Specifically, we have redrawn Figure 2A to depict the uniquely mapped and multi-mapped reads more clearly. Additionally, we have included pi and theta in Figure 2C as requested. We acknowledge the concern about the labeling of "Read 1" and have taken steps to address this by removing it and providing a more accurate representation in Figure 2B.

Figure 2: Schematic overview of LocusMasterTE

An example illustration for the LocusMasterTE pipeline is presented. (A) The short-read RNA sequencing data is aligned using the provided annotation file, and reads are classified into uniquely mapped and multi-mapped reads based on the alignment score. Uniquely mapped reads and multi-mapped reads are then aligned to TE A, TE B, and TE C. Long-read RNA sequencing, preferably ONT, is aligned using minimap2 [23], and TEs are quantified using featureCounts [18] with the same annotation file provided for short-read RNA sequencing. TPM counts for long-reads are calculated, and the belonging subfamily is provided for each individual TE. (B) Initially, equal weights are assigned to all three TEs - TE A, TE B, and TE C (TE A:0.33, TE B: 0.33, TE C: 0.33). (C) Two critical parameters, pi (π) and theta (θ), represent the proportion of fragments originating from the transcript and the proportion of non-unique fragments to be reassigned, respectively. These parameters are updated with TPM fractional values from long-read RNA sequencing. During the E-step, MLE is calculated for all TEs. In the M-step, MAP estimates are updated. (D) Finally, after the EM algorithm coverages, best-aligned read counts are determined, leading to the final assignment of TE counts.

- Supplementary Figure 3 is missing in the main manuscripts, there is only the title and the legend (p46).

We thank the reviewer and supplementary figure 3 (now supplementary figure 4 in revised version) has been added and referred to in text.

Page 5, line 148-149:

Prior to comparing TE counts, we tested the quality of simulated short-read using coding genes (Supplementary Fig. 4).

- in section 2.4, again, concerning the TE age, it would be better to have %divergence in addition to the statistical test names and associated p-values (that should be corrected for multiple testing). It should also be mentioned in the text that fig4B has been done using HCT1116 data. I am also wondering why the R2 for this tissue is different in the figure 4E. The increase in correlation is not very high. It is possible to estimate the added-value by comparing both trends statistically? It seems less efficient than compared to the simulation.

We thank the reviewer for this comment. First, we used %divergence as a representation of TE age. Moreover, we added information about statistical test names, paired t-tests in the figure legend.

Figure 4A: Age comparison (%divergence) between two groups of TEs. Paired t-test was conducted in every cell line samples.

Figure 4A legend:

Age distribution of TEs across all cell lines between two groups of TEs. Red: TEs quantified differently between LocusMasterTE and Telescope [8] (TEs corrected by LocusMasterTE). Blue: TEs quantified the same across LocusMasterTE and Telescope [8] (TEs initially correct). We conducted a paired t-test for each cell line sample. Quantitative numbers are labeled above each box.

The reason for the difference between the correlation in 4B and 4E is because 4B uses our in-house HCT116 data which has substantially higher sequencing depth (avg (depth) = 16.6078) compared with the HCT116 data from SG-NEx in 4E (avg (depth) = 8.5334). This information has now been

included in Supplementary Figure 7.

Supplementary Figure 6: Average depth across all 6 samples.

Thank you for the suggestion. We have now also performed paired t-test between correlation between long-read and Telescope vs correlation between long-read and LocusMasterTE (p-value=0.0062; mean(Blue): 0.4740 mean(Red): 0.5200).

Figure 4C: Summary bubble plot for correlation with long-read in SG-NEEx cell lines. Paired t-test was conducted

7. I am quite puzzled by the results in section 2.5. What about polymorphic TE insertions? All RNA-seq mappings are done on a reference genome, with a particular set of TE insertions. However the RNAs come from other individuals which will have different TE insertions, especially the most recent ones. Then, I am wondering how this may affect the results. Some of the reads (either short or long) will come from insertions that may not be in the reference genome. I am not sure what the authors want to show here. The TE expression correlation and correlation of gene expression do not prove anything.

We thank the reviewer for their comment.

LocusMasterTE is primarily designed to detect RNA from TEs annotated in the reference genome. Therefore, TE insertions not included in the reference genome would not be quantified. These missed TE insertions can be quantified by performing *de novo* transcriptome assembly using Trinity, followed by creating a custom STAR index. RepeatMasker would then identify TE-derived transcripts.

We applied this approach in our in-house HCT116 cells to observe differences between reference genome-aligned TE counts and *de novo* transcriptome-aligned TE counts. We generated a correlation plot between the transcriptome-aligned and reference genome-aligned data. Additionally, we selected young TEs (mean % divergence < 20.0%) to draw a correlation plot, further validating our approach.

However, creating *de novo* transcriptomes for all TCGA-COAD samples is challenging. As we observed a good correlation between reference genome-aligned TE counts and *de novo* transcriptome-aligned TE counts, we decided to align to the reference genome for the rest of the analysis in the TCGA-COAD section.

Moreover, as the Editor suggested that our manuscript aligns better with a Methods paper, we have reorganized the TCGA-COAD section, removing the gene and TE correlation and survival analysis parts. Thus, an analysis quantifying somatic TE insertions is out of scope of our current manuscript.

Response Figure 1: Correlation quantification of TE subfamilies between reference genome (hg38) aligned vs De Novo Transcriptome aligned in in-house HCT116. (A) all TE subfamilies (B) young (mean %divergence < 20.0%) TE subfamilies.

8. The method section is not very clearly organized. I would suggest a complete reorganization that would help the reader follow exactly what has been done. Moreover, some titles are very not informative (like for example the section 4.6).

We appreciate the feedback from the reviewer. We reorganized the method section. First we started by describing the datasets used in our study at the beginning of the Methods section. All information related to the generation of datasets were followed. Additionally, we have replaced some titles with better ones. Specifically, section 4.6 (which no longer has a section number) has been modified to:

Page 11, line 361:

Comparative analysis on proportion of multi-mapping reads

9. about the option concerning the weight that can be associated to long-reads in the model. I think this deserves to be better explain, especially, what are the possible choice of the user, what they mean, what are their limits etc. It will be difficult to always find both short and long reads obtained from the same dataset which may really be a problem. I would like to have more information and test about what is happening if both data are not exactly coming from the same source.

We thank the reviewer for this suggestion. As pointed out by the above comment, it is hard to find a match between short and long-read. In order to tackle this, (`--long_read_weight`) is introduced. We used SG-NEx study K562 cell line as short-read RNA-seq and input in-house HCT116 Nanopore RNA-seq as long-read input and tested with a series of values ranging from 0.01 to 1.5. The correlation test was performed between K562 Nanopore RNA-seq and quantified values.

Supplementary 3: Impact of (`--long_read_weights`) in quantified TE of LocusMasterTE

The description of (`--long_read_weights`) results is as follows:

Page 4, line 123-135:

Although data from any long-read sample can be used in LocusMasterTE, the best accuracy would be achieved using matched cell or tissue types between long-read and short-read. Nevertheless, we recognize that long-read data may not always be available for the desired cell type, thus, LocusMasterTE introduces a weight parameter (`--long_read_weight`), allowing users to manage the contribution of long-read information. Lower weights are recommended with distant tissue types, but optimal weights vary between samples. We conducted tests with various weight (`--long_read_weight`) values from 0.01 to 1.5 on our in-house HCT116 short-read cell line sample along with a K562 long-read sample from the SG-NEx study [18]. Our findings revealed that values below 0.5 and above 1.0 (`--long_read_weight`) resulted in lower accuracy (Supplementary Fig. 3). When an unmatched long-read sample is used, it is advisable to use a weight (`--`

long_read_weight) smaller than 1.0. Otherwise, in most cases, a weight (--long_read_weight) of 1.0 is recommended (default).

10. section 4.4, p19, line 9: simple repeats are not transposable elements. There is no real point to mention them.

We thank the reviewer and remove sentences with simple repeats.

11. section 4.4: the age analysis needs to be performed differently. It is not clear what is the "milliDiv" column present in the repeatmasker file (not database). And what file exactly. I do not recall that this column exists in the .out output file. As I specified before, age should be estimated as %divergence or using Kimura distance metrics.

We thank the reviewer for this comment and change to %divergence from the RepeatMasker track. Also further clearly mentioned the source of % divergence in previous method 4.4 (now does not have section number):

Page 12, line 409-410:

We acquired % divergence (% divergence from ancestral sequences) of TE individuals from the RepeatMasker table at the UCSC database.

12. in section 4.10.2 about histone enrichment analysis. This section should be more detailed, explaining the type of information that can be found in bigwig files, what scores were calculated exactly and how it is possible to reconcile results from chip-seq that comes from short-reads with the same multi-mapping problems as RNA-seq short-read when it comes to TEs.

Thank you for the comment. In response to the reviewer's feedback, we have provided additional information about histone bigwig files in previous section 4.10.2 (now does not have section number) and in the Figure 4 legend.

Figure 4 Legend:

(D) Two repressive chromatin fold change over control coverage of in-house HCT116 cell sample for the same two TE groups. Brown: TEs quantified higher in LocusMasterTE (TEs measured highly by LocusMasterTE). Blue: TEs are quantified higher in Telescope [8] (TEs are measured highly by Telescope). (E) Three active chromatin fold change over control coverage of in-house HCT116 cell line for two TE groups.

We agree that short-read chip-seq has a multi-mapping issue. Therefore, we investigated the mappability of two groups (TEs with higher expression by LocusMasterTE or Telescope) in in-house HCT116 sample. Upon comparing the distribution of mappability of the two TE groups, is similar genome mappability and thus unlikely to greatly affect our comparison of histone profiles in Figure 4D-E

Supplementary figure 8A: Distribution graph of average mappability in two groups of TEs.

Page 6, line 197-202:

To avoid bias in the histone ChIP-seq data, we selected TE regions with a mappability score greater than 0.25, finding no significant difference between the two groups (median TEs with higher expression by Telescope) = 0.5006275 vs median (TEs with higher expression by LocusMasterTE) = 0.505051, $p = 0.1661$, Wilcoxon signed-rank test, Supplementary Fig. 8A).

- it would be nice to have the same figure as figure 1C with the results provided by LocusMaster to see how the % of cases where the EM is not able to assess unique matches evolves.

We thank the reviewer for this suggestion and add figure for LocusMasterTE as follows.

Supplementary Figure 2: A comparison between Telescope and LocusMasterTE in the proportion of cases where EM successfully identifies the best hit

Page 4, line 120-122:

The occurrences in which the EM algorithm failed to identify a single locus were reduced to 1% with the use of LocusMasterTE (Supplementary Fig. 2), indicating the benefit of incorporating long-read RNA-seq information.

14. P3: the « Endogenous » term should be removed since it is redundant with the term Transposable elements which are by definition endogenous.

We thank the reviewer for this correction and remove endogenous term in start of abstract and introduction.

Page 1, line 10-12, abstract:

Transposable elements (TEs) can influence human diseases by disrupting genome integrity, yet their quantification has been challenging due to the repetitive nature of these sequences across the genome.

Page 1, line 23-24, introduction:

Transposable elements (TEs) are repetitive genomic units accounting for half of the human genome [1].

15. there is a repeat in the second sentence of the introduction p3 (« Based on genetic etc. »).

We thank the reviewer for pointing out this. We have rephrased repetitive

sentences. Page 1, line 24-25:

TEs can also be grouped based on genetic composition and monophyletic origin: class > family > subfamily [1].

16. the authors indicate that "most TEs are now inactive at the transcriptional level" (p3, line25). But it actually depends on the species considered. They should precise that it is the case in the human genome.

We thank the reviewer and specify human in the

sentence. Page 1, line 28-31:

With the accumulation of mutations through evolution [4], most TEs are now inactive at the transcriptional level in humans; however, recent studies have identified TEs with functional impact on health and disease [5].

17. p6 line 51 coordinations => coordinates

We thank the reviewer for correction and

revised it. Page 3, line 104-105:

As LocusMasterTE was built to quantify individual copies of TEs, a GTF containing the coordinates of individual TE copies is required.

18. p13 line 8: TEs do not "impose" essential roles. I would modify this sentence.

We thank the reviewer for this and re-write sentence as below:

Page 7, line 227-228:

Transposable elements play essential roles in cancer and complex biological diseases in humans [1].

19. there are many typos (to numerous to be point out individually) in the whole manuscript that need to be corrected

We appreciate the reviewer and corrected typos throughout manuscript.

Additional Changes

1. To emphasize the importance of the 6% failure rate in EM's best hit conclusion, we also explore the age difference between the two groups.

Fig 1D. %divergence in EM has more than 2 best hits (Brown) vs EM successfully conclude best hit (Green)

2. We have used in-house HCT116 with deeper depth. Thus, all the figures created based on in-house HCT116 have been revised.
3. As the editor suggested that our manuscript should be included in the "Methods" paper, we reorganized it to follow the Genome Biology "Methods" paper organization, including reducing the abstract to 100 words and adding a conclusion.
4. We removed section numbers throughout the manuscript.

Second round of review

Reviewer 2

Globally the authors have made a quite good job in this revised manuscript. It is much more clear to follow, especially the methodology. I find the figure 2 more straightforward and understandable.

I thus appreciate the effort of the authors to take into account the majority of my remarks. There are still some points that need improvement, especially on the statistical point of view.

1) The authors have mainly used the student t-test to compare the %divergence (figure 1D and E) but I am not quite sure this is relevant since it is a parametric test assuming the data distribution is normal. Unless this was indeed checked (and thus it should be indicated) I would rather use the equivalent non-parametric test, although it will probably not change the conclusions.

2) On Figure 1A, a test has been performed since on the figure a star is indicated but there is no mention of the test nor the p_value, nor in the text or in the legend.

3) Several correlations have been made with the mention of the R2 but without indicating the p_value (in the text) nor the type of test (pearson or spearman? The last would be probably more appropriate).

4) In the result part, line 191, R2 correlation values are mentioned (0.39 and 0.35) but they are not the values that appear on figure 4B nor in the figure 4C.

5) As for my comment regarding the comparison of the correlations obtained for the results from Telescope and those from LocusMasterTE. I am again not quite sure that it is relevant to globally compare the correlations with a paired t-test. I was rather thinking about separate tests for each tissue (with a multiple test correction to adjust to obtained p_values) using the Fisher r-to-z transformation.

Authors' response to reviewers

1. The authors have mainly used the student t-test to compare the %divergence (figure 1D and E) but I am not quite sure this is relevant since it is a parametric test assuming the data distribution is normal. Unless this was indeed checked (and thus it should be indicated) I would rather use the equivalent non-parametric test, although it will probably not change the conclusions.

Thank you for pointing this out. We agree that a non-parametric test is more appropriate than the Student's t-test in this case. Consequently, we have performed a Wilcoxon signed-rank test

instead and now include the corresponding p-value in the text. Page 3, line 83-93:

The estimated age (quantified by % divergence from a consensus ancestral sequence) of these ambiguously assigned TEs is significantly younger (i.e. lower % divergence) than other quantified (14.5% versus 19.0%; Wilcoxon signed rank test, $p < 2.2e-16$, Fig. 1D), suggesting that existing EM-based methods are more prone to misquantify younger TEs.

Given that the longer read length in long-read sequencing leads to lower multi-mapping rates, we observed that the divergence of TEs quantified by long-read was lower than those captured by short-read (15.8% vs 18.7%; Wilcoxon signed rank test, $p < 2.2e-16$). The difference was increased when we compared a group of TEs exclusively quantified by long-read (median = 15.2%) and TEs only quantified by short-read (median = 19.5%; Wilcoxon signed rank test, $p < 2.2e-16$) (Fig. 1E).

2. On Figure 1A, a test has been performed since on the figure a star is indicated but there is no mention of the test nor the p_value, nor in the text or in the legend.

Thank you. We have now clearly indicated in both the text and figure legend that a Chi-square test was performed, and we have included the corresponding p-value.

Legend of Fig 1A. Percentages for uniquely mapped reads and multi-mapped reads for short- read and long-read measured by BWA. Significance was evaluated by Chi-squared test.

Page 2, line 61-63:

This greatly decreases the proportion of multi-mapping reads during alignment to TE in transcriptomics data (Chi-squared test, $p = 0.003891$, Fig. 1A).

3. Several correlations have been made with the mention of the R2 but without indicating the p_value (in the text) nor the type of test (pearson or spearman? The last would be probably more appropriate).

Thanks for raising this issue. We have now clarified in the text that we performed a Spearman correlation test instead of Pearson correlation test. Additionally, we have now included the corresponding p-value in the text to provide a complete description of the statistical analysis. All figures that included correlation analyses have been updated accordingly.

Page 2, line 67-72:

As TE subfamily is an aggregation of individual copies, we observed good agreement between Oxford Nanopore Technologies (ONT) long-read and short-read (Spearman's rank correlation coefficient, $R^2=0.84$, $p < 2.2e-16$, Fig. 1B); however, there was a more substantial difference at locus-specific copy level (Spearman's rank correlation coefficient, $R^2=0.32$, $p < 2.2e-16$). For comparison, the correlation at coding genes is $R^2=0.57$ (Spearman's rank correlation coefficient, $p < 2.2e-16$, Supplementary Fig. 1A).

Page 6, line 186-189:

We observed an increased correlation in TE expression with long-read results using LocusMasterTE (Spearman's rank correlation coefficient, $R^2 = 0.44$, $p < 2.2e-16$) compared with Telescope (Spearman's rank correlation coefficient, $R^2 = 0.41$, $p < 2.2e-16$) using the inhouse HCT116 dataset [8] (Fig. 4B).

Legend of Fig 1B. Correlation plots between log10 TPM count for long-read and short-read in two same domains.

Legend of Fig 4B. Spearman correlation plots for Telescope [8] and LocusMasterTE with long-read inhouse HCT116 cell line. log10 TPM counts are used.

Legend of Supplementary Figure 1: (A) Comparison of coding genes captured by short-read and long-read RNA-seq. Venn diagram showing the number of captured genes in two RNA-seq techniques. Spearman correlation plot between TPM count for short-read and long-reads in the coding genes domain. (B) Comparison between downsampled short-read and long-read RNA-seq. The short-read sample is downsampled to match the number of bases mapped between short-read and long-read. Venn diagrams for downsampled short-read and long-read of TEs in subfamily and individual levels. Spearman correlation plots between TPM count for downsampled short-read and long-reads in the same domains.

S7

Supplementary Figure 7: Series of Spearman correlation plots between long-read and short-read for 5 SG-NEx cell lines. Telescope and LocusMasterTE quantify TEs in 5 cell line samples. Generated TE matrices are compared with the long-read result. Brown: TEs quantified higher in LocusMasterTE (TEs measured highly by LocusMasterTE). Blue: TEs are quantified higher in Telescope [8] (TEs are measured highly by Telescope).

4. In the result part, line 191, R2 correlation values are mentioned (0.39 and 0.35) but they are not the values that appear on figure 4B nor in the figure 4C.

We apologise for incorrectly reporting R^2 values in the text. We have now fixed this error by replacing the R^2 values to the correct ones from the Fig 4B.

Page 6, line 186-189:

We observed an increased correlation in TE expression with long-read results using LocusMasterTE (Spearman's rank correlation coefficient, $R^2 = 0.44$, $p < 2.2e-16$) compared with Telescope (Spearman's rank correlation coefficient, $R^2 = 0.41$, $p < 2.2e-16$) using the inhouse HCT116 dataset [8] (Fig. 4B).

5. As for my comment regarding the comparison of the correlations obtained for the results from Telescope and those from LocusMasterTE. I am again not quite sure that it is relevant to globally compare the correlations with a paired t-test. I was rather thinking about separate tests for each tissue (with a multiple test correction to adjust to obtained p_values) using the Fisher r-to-z transformation.

Thank you for this excellent suggestion, and apologies for missing this in our previous revision. We have performed Fisher's r-to-z transformation and added Bonferroni-corrected p-values for each cell line sample. All p-values have been clearly indicated in the revised manuscript.

Legend of Fig 4C. Summary bubble plot for correlation with long-read in SG-NEx cell lines. Fisher r-to-z transformation was performed between two groups with Bonferroni correction for multiple testing applied.

Page 6 line 191-195:

Nevertheless, the same trend was consistently observable in SG-NEx across all cell types. (Fisher r-to-z transformation, Bonferroni correction, p -adj A549 = 0.0392, p -adj HepG2 = 0.0301, p -adj HCT116 = 0.0001, p -adj K562 = 0.0016, p -adj MCF-7 = 0.0314, Fig. 4C; Supplementary Fig. 7).

Additional Changes

At the end of each figure legend containing *, we have added the following significance level notation: (ns: $p > 0.05$, *: $p \leq 0.05$, **: $p \leq 0.01$,

Third round of review

Reviewer 2

The authors have satisfactorily answered my concerns regarding my last remarks.